# *Cinnamomum tamala* Leaf Extract Stabilized Zinc Oxide Nanoparticles: A Promising Photocatalyst for Methylene Blue Degradation

**DOI:** 10.3390/nano11061558

**Published:** 2021-06-13

**Authors:** Sajina Narath, Supin Karonnan Koroth, Sarojini Sharath Shankar, Bini George, Vasundhara Mutta, Stanisław Wacławek, Miroslav Černík, Vinod Vellora Thekkae Padil, Rajender S. Varma

**Affiliations:** 1Department of Chemistry, School of Physical Sciences, Central University of Kerala, Kasaragod 671316, Kerala, India; sajina@cukerala.ac.in (S.N.); supinkk09@gmail.com (S.K.K.); 2Department of Biochemistry and Molecular Biology, School of Biological Sciences, Central University of Kerala, Kasaragod 671316, Kerala, India; 3Department of Medicine, Thomas Jefferson University, Jefferson Alumni Hall, 1020 Locust Street, Philadelphia, PA 19107, USA; 4Polymer and Functional Materials Department, CSIR-Indian Institute of Chemical Technology, Tarnaka 500007, Hyderabad, India; mvas@iict.res.in; 5Institute for Nanomaterials, Advanced Technologies and Innovation (CXI), Technical University of Liberec (TUL), Studentská 1402/2, 461 17 Liberec, Czech Republic; 6Regional Centre of Advanced Technologies and Materials, Czech Advanced Technology and Research Institute, Palacky University, Šlechtitelů 27, 783 71 Olomouc, Czech Republic

**Keywords:** green synthesis, zinc oxide nanoparticles, photocatalysis, methylene blue

## Abstract

A facile green synthetic method is proposed for the synthesis of zinc oxide nanoparticles (ZnO NPs) using the bio-template *Cinnamomum tamala* (*C. tamala*) leaves extract. The morphological, functional, and structural characterization of synthesized ZnO NPs were studied by adopting different techniques such as energy dispersive X-ray analysis (EDX), high-resolution transmission electron microscopy (HR-TEM), scanning electron microscopy (SEM), X-ray diffraction (XRD), UV-Visible spectroscopy, fourier transform infrared (FTIR) spectroscopy, raman spectroscopy, and X-ray photoelectron spectroscopy (XPS). The fabricated ZnO NPs exhibit an average size of 35 nm, with a hexagonal nanostructure. Further, the well-characterized ZnO NPs were employed for the photocatalytic degradation of methylene blue (MB) in an aqueous solution. The photocatalytic activity was analyzed by changing the various physicochemical factors such as reaction time, amount of photocatalyst, precursor concentration, and calcination temperature of the ZnO NPs. All the studies suggest that the ZnO synthesized through the green protocol exhibits excellent photocatalytic potency against the dye molecules.

## 1. Introduction

Environmental nanotechnology has garnered immense significance for removing hazardous chemicals, with remarkable potential to promote economically viable synthesis. Nanoparticles (NPs), the building blocks of nanotechnology, are categorized into different classes based on their morphology, structure, size, and chemical properties [1,2]. The synthesis of metal oxide NPs has been widely employed using physical and chemical methods, including electro-explosion, solvothermal, hydrothermal, microwave, spray pyrolysis, vapour deposition, microemulsion, coprecipitation, and the wet-chemical method [3,4,5,6]. However, these conventional methods of the synthesis of NPs limit their application due to the issues of toxicity, costliness, and the formation of diverse nanostructures [3]. In recent times, the emergence of various synthesis methods for semiconductor nanoparticles gained much attention [7,8]. The synthesis of NPs using green chemistry principles provides a solution to ecological concerns by minimizing toxic chemicals and the subsequent conversion of products without much process investment [9,10]. The greener approach to the synthesis of NPs came into the limelight to restrain the ecosystem with a naturally available biodegradable matter for its production [11,12]. The plant extract-mediated synthesis is a simple, cost-effective, efficient, and feasible process because of the availability of effective phytochemicals and functional groups present in the extractives [13].

*C. tamala* is an evergreen tree belonging to the Lauraceae family, commonly known as Tejpat in India [14]. It is medicinally used as a carminative, anti-flatulent, diuretic, and for cardiac disorders. Alkaloids, terpenes, flavonoids, tannins, polyphenols, and saponins are present in *C. tamala* [15], which can facilitate the synthesis of nanoparticles by serving as a reducing, as well as a capping or stabilizing, agent.

Zinc oxide (ZnO) is a promising metal oxide that can eliminate most of the ecological concerns, as it is relatively nontoxic [16,17,18] and can be used for the removal of environmental pollutants via photodegradation (especially wastewater treatment); ZnO NPs have acquired attention due to their extraordinary property of degrading contaminants [19]. One of the most important applications of semiconductor nanoparticles is their photocatalytic activity, which is crucial in removing or degrading pollutants. The industrially essential organic dyes extensively deployed in textile industries releases nearly 146,000 tonnes of dyes per year, alongside other associated pollutants [1,20,21]. The large number of dyes discharged in water bodies are difficult to degrade because of their non-biodegradable nature, large size, and complex nature; they often threaten humans and the environment [22]. The commonly used techniques are efficient in degradation, but they cause the generation of absorbent material, thus needing the post-treatment of waste materials [21,23,24,25]. Extensive research has been conducted to replace these methods, and semiconductor photocatalysis is emerging as the best choice for this purpose, as it has remarkable potential to destroy many toxic compounds in an aqueous medium, with fewer or no harmful products. The solar-driven photocatalytic processes has recently gained special attention in solving wastewater problems, as it relies on a natural process of energy conversion [26].

The present study has been focused on the fabrication of green ZnO NPs using *C. tamala* leaf extract and evaluating its ability towards methylene blue (MB) photodegradation. The photocatalytic potency or effective degradation of MB dye was studied using various physicochemical parameters such as reaction time, the concentration of precursor, calcination temperature, and amount of catalyst.

## 2. Materials and Methods

### 2.1. Materials

Zinc nitrate hexahydrate (98% purity) and methylene blue were procured from Sigma Aldrich and Himedia chemicals, Mumbai, India. *C. tamala* leaves were collected from the Kannur district, Kerala, India. All other chemicals and solvents used were of analytical grade and were used without further purification.

### 2.2. Preparation of C. tamala Aqueous Extract

Fresh *C. tamala* leaves were collected, cleaned well, and shade dried for a month. The wholly dried leaves were ground to a fine powder using a mixer grinder.

Ten g of fine powder was weighed and transferred to a 500 mL stoppered bottle. Two hundred and fifty mL of distilled water was added, and the mixture was heated at 80 °C on a magnetic stirrer for 1 h. The temperature was maintained constantly and, subsequently, the solution was filtered using a cotton cloth. The filtrate solution was used for the synthesis of ZnO NPs, and it was stored in the refrigerator for further use.

### 2.3. Synthesis of Zinc Oxide Nanoparticles (ZnO NPs)

ZnO NPs were synthesized using *C. tamala* extract by a simple procedure. Briefly, *C. tamala* extract (5 mL) was added to 0.1 M of zinc nitrate (Zn(NO_3_)_2_·6H_2_O) (25 mL) and stirred at 60 °C for 3 h. The product obtained was dried and calcined at 500 °C for 2 h.

### 2.4. Characterization of ZnO NPs

#### 2.4.1. X-ray Diffraction (XRD) Analysis

To study the crystal structure of ZnO NPs, the calcined sample was subjected to X-ray diffraction analysis with a Rigaku Miniflex 600 (Tokyo, Japan) diffractometer with nickel filtered Cu Kα (λ = 1.54 Å) radiation. The sample scanned between 20° and 90° (diffraction angle 2θ) with a scan speed of 0.05°. The Joint Committee on Powder Diffraction Standards (JCPDS)’s data files were used for matching with the spectrum obtained from ZnO NPs.

#### 2.4.2. FT-IR Analysis

Fourier Transform Infrared analysis was performed using a Perkin-Elmer FTIR Spectrum Two spectrometer (Singapore, Singapore) in attenuated total reflection (ATR) mode with 30 scans in a spectral range set between 4000 and 400 cm^−1^.

#### 2.4.3. UV-Visible Analysis

Optical properties were obtained using an ultraviolet (UV)-visible spectrophotometer (Perkin Elmer Lambda 35, Singapore) over the spectral region 200–700 nm. The present spectrophotometer uses a deuterium lamp for UV and a tungsten lamp for the visible region.

#### 2.4.4. Scanning Electron Microscope-Energy-Dispersive X-ray Spectroscopy (SEM-EDX) Analysis

The synthesized ZnO NPs were analyzed with a scanning electron microscope (TECNAI), Model: AIS2100 from M/S. Mirero Inc, Seongam, Korea, to determine the surface morphology. The elemental composition of the sample was acquired from energy-dispersive X-ray spectroscopy (EDX) measurements, carried out using a scanning electron microscope equipped with an EDX: Model No. INCAE350 from M/S, Oxford Instruments, Abingdon, UK.

#### 2.4.5. TEM and High-Resolution Transmission Electron Microscopy (HR-TEM) Analysis

The nanoparticles size and particle distributions were attained from transmission electron microscopy analysis (TEM) (JEOL, JEM-2100, Tokyo, Japan).

#### 2.4.6. Raman Spectroscopy

The Micro Raman scattering study was performed using LabRam HR Evolution Raman Spectrometer (Horiba Scientific, France with 633 nm Laser in backscattering geometry).

#### 2.4.7. XPS

The Thermo Scientific K-Alpha instrument (monochromatic Al Kα radiation, Ephoton = 1486.6 eV) was used for X-ray photoelectron spectroscopy (XPS) studies. The internal reference peak of C 1s peak centred at 284.8 eV is used for binding energy (B.E) corrections. A dual-beam charge neutralization method is used in the instrument, where a low energy ion beam is used to eliminate the samples static charge, allowing the low energy electron beam to reach the sample and neutralize the localized positive charge created by the X-ray beam. The specimens were analyzed at an electron take-off angle of 70°, and measured with respect to the surface plane. The monochromatic X-ray source is located perpendicular to the analyzer axis, and the standard X-ray source is located at 54.7° relative to the analyzer axis. For all of the spectra, the spectrometer was operated in a standard mode. All survey spectra scans were taken at a pass energy of 58.7 eV. The narrow scans of strong lines were, in most cases, just wide enough to encompass the peaks of interest and were obtained with a pass energy of 23.5 eV.

### 2.5. Photocatalytic Studies on Methylene Blue Degradation

The photocatalytic activity of green synthesized ZnO NPs was evaluated by MB degradation under direct sunlight. The experiment setup is made in such a way that the reaction mixture was exposed to solar radiation. The intensity of solar light was measured using a lux meter. The degrading capacity was assessed by taking out 5 mL of the reaction mixture at a regular interval of time. The samples were centrifuged at 9000 rpm for 10 min and were analyzed for the photocatalytic degradation of MB using a UV-visible spectrometer over the wavelength 200–700 nm. The percent of photodegradation of MB (photocatalytic efficiency of ZnO) in aqueous media was calculated by the following equations: η = (C_0_ − C)/C_0_ or η = (A_0_ − A)/A_0_ [21], where C_0_ is the initial dye concentration without catalyst and C is the final dye concentration with catalyst after 90 min, and A_0_ and A show initial and final absorbance of MB dye without and with ZnO photocatalyst.

#### 2.5.1. Effect of Solar Irradiation Time on Photocatalytic Studies

To evaluate the influence of solar irradiation time on MB degradation, 5 mg of green ZnO NPs was added to 50 mL of dye solution. The resultant reaction mixture was kept for continuous stirring at 700 rpm under solar irradiation. The percentage degradation of MB was evaluated by the absorbance measurements of the solution using a UV-visible spectrophotometer by eluting and measuring the absorbance of 5 mL supernatant (centrifuged at 9000 rpm for 10 min) at regular time intervals of 15 min. The process was continued for 90 min.

#### 2.5.2. Effect of Amount of Catalyst on Photocatalytic Studies

The influence of the amount of catalyst on photocatalytic degradation of MB was evaluated by adding different amounts (5–15 mg) of ZnO NPs to 50 mL of MB dye solution (10 µM). The reaction mixture was kept for stirring on a magnetic stirrer for 90 min under sunlight. The absorbance measurements were done with a UV-visible spectrometer and calculated the photodegradation capability of ZnO NPs.

#### 2.5.3. Effect of Calcination Temperature on Photocatalytic Studies

Green synthesized ZnO NPs calcined at three different temperatures (300 °C, 500 °C, and 700 °C) was used for MB photodegradation to study the influence of calcination temperature of the sample on photocatalytic efficiency. Five mg of each of them was added to 50 mL of MB dye solution. The reaction mixture was kept on a magnetic stirrer for 90 min under sunlight. The supernatant solution was subjected to absorbance measurements.

#### 2.5.4. Effect of Concentration of Zn(NO_3_)_2_ on Photocatalytic Studies

To study the effect of precursor concentration on photocatalytic degradation capability, ZnO NPs were synthesized from three different precursor concentrations, i.e., 0.025, 0.05, and 0.1 M of Zn(NO_3_)_2_·6H_2_O. Synthesized nanoparticles were calcined at 500 °C. The photocatalytic study was monitored by adding 5 mg of catalyst to 50 mL MB dye solution using a UV-visible spectrophotometer.

### 2.6. Effect of Solar Irradiation on Photodegradation of MB

The effect of solar irradiation on the percentage degradation of MB dye solution was evaluated by loading ZnO catalyst in the presence and absence of sunlight. Fifteen mg of ZnO NPs was added to 50 mL of MB dye solution, and the sample was mixed thoroughly for 90 min on a magnetic stirrer. The reaction was performed under sunlight and in a dark chamber to study the effect of solar irradiation on MB degradation.

### 2.7. Studies on the Effect of Concentration of Zn(NO_3_)_2_ on Crystallite Size of ZnO NPs

To study the relationship between the concentration of precursor and the crystallite size, ZnO NPs were synthesized from 0.025, 0.05, and 0.1 M Zn(NO_3_)_2_·6H_2_O. All samples were calcined at 500 °C. From the XRD spectrum obtained, the crystallite size of NPs can be calculated using Debye–Scherrer’s Equation:D = 0.9λ/(βcosθ)
where D is the crystallite size, λ is the X-ray wavelength, θ is Bragg’s angle in radians, and β is the full width half maximum [27].

## 3. Results and Discussion

### 3.1. Mechanism of Formation of ZnO NPs via Biosynthesis

The environment is rich in plant resources which contain many phytochemicals present in roots, leaves, flowers, etc. The functional groups, such as hydroxyl, carbonyl, carboxylic acid, etc., present in these phytochemicals can facilitate the formation of nanoparticles by inducing chemical reduction. The current work is dealing with biosynthesizing ZnO NPs using a green template, *C. tamala* leaf extract, which is a rich source of polyphenols, flavonoids, and alkaloids [15].

When a metal solution, i.e., Zn(NO_3_)_2_·6H_2_O, is introduced into a well synthesized homogenous leaf extract solution, leaf extract will try to form the matrix in which zinc in the +2 oxidation state gets absorbed. The stability of Zn^2+^ on the matrix is due to the chelating effect, or a type of interaction between functional groups and Zn^2+,^, which may be due to the transfer of a lone pair of electrons (available on functional groups) to the empty orbital of zinc. Therefore, the quarantined cations M^n+^ or hydroxylated cation [M(OH)]^m+^ can go through nucleation of the growth process and are accelerated by functional groups present in the leaf extract [26]. The dried mixture obtained was yellow-coloured, which may be attributed to the phytochemicals present. The sample was kept for calcination to remove the hydroxides and other impurities that rise to the formation of required oxide [28] and, finally, results in ZnO NPs. Various analytical techniques confirmed the formation of nanoparticles.

### 3.2. Characterization of ZnO NPs

#### 3.2.1. X-ray Diffraction Studies

The X-ray diffraction pattern of ZnO NPs calcined at 500 °C is represented in Figure 1. A definite number of sharp, intense diffraction peaks were observed, corresponding to (100), (002), (101), (102), (110), (103), (200), (112), and (201) crystallographic planes at characteristic peak angles 31.6°, 34.31°, 36.15°, 47.44°, 56.51°, 62.81°, 66.36°, 67.85° and 69.02°, which have been in agreement with the hexagonal wurtzite crystalline structure of ZnO NPs as per Joint Committee on Powder Diffraction Studies Standards (JCPDS card no. 36–1451) [27,28], thus validating the formation of required ZnO NPs. The intense and narrow peaks obtained described the crystalline nature and high purity of ZnO. In contrast, the absence of additional peaks further confirms the higher purity of ZnO NPs, since they are free of impurities. The Debye–Scherrer equation is employed to find out the average crystallite size or diameter of green synthesized ZnO NPs, D = 0.9λ/(βcosθ), corresponding to the maximum intense peak of the (101) plane. The average crystallite size is estimated to be approximately 35 nm.

#### 3.2.2. FTIR Analysis

FTIR spectroscopy was employed to recognize the functionalities present in the *C. tamala* leaf extract and characterize the ZnO NPs, and the obtained spectra are presented in Figure 2a. In the mid-IR region, peaks reveal the comparison of two plots that guide the confirmation of ZnO NPs. The leaf extract spectrum was observed to have significant peaks at 3500–3100 cm^−1^, 2924 cm^−1^, 1603 cm^−1^, and 1021 cm^−1^. The broad peak obtained at 3500–3100 cm^−1^ was attributed to the OH stretching frequency. Similarly, the bands registered at 2924 cm^−1^, 1603 cm^−1^, and 1021 cm^−1^ were due to the C–H, C=C, and C–O stretching vibrations, respectively [29]. These peaks shown by the leaf extract were due to the phytochemicals present in it, i.e., glycosides, flavonoids, polyphenols, terpenoids, tannins, carbohydrates, and reducing sugar. These phytochemicals have shown other additional small peaks that are due to overtones. The absence of these specified peaks in the FTIR spectrum of ZnO nanopowder results from the higher purity of green synthesized ZnO NPs calcined at 500 °C. However, the ZnO NPs showed a unique sharp signal in the fingerprint region below 1000 cm^−1^, which may be attributed to the stretching frequency of the Zn–O bond [30]. Besides, ZnO NPs also showed small intense peaks which may be corresponding to the complication of fundamental bands, i.e., overtones, combination bands, or different bands.

An absorption peak corresponding to the O–H frequency is barely seen in ZnO NPs. However, a broad peak at 3293 cm^−1^ is noticed in the enlarged view of the figure, as depicted in Figure 2b, suggesting the O–H band is present in the ZnO sample. The FT-IR analysis manifests the role of organic substances present in *C. tamala* leaf extract to stabilize biosynthesized ZnO NPs. Thus, the formation of ZnO NPs is accomplished due to the interaction between oxygen in functional groups involved in *C. tamala* leaf extract and zinc molecules in the precursors.

#### 3.2.3. Optical Property Studies

The bio-synthesized ZnO NPs were further analyzed using a UV-visible spectrophotometer in the range of 200–700 nm, and the resulting optical plot is shown in Figure 3a,b. The observed absorption spectrum of synthesized nanoparticles matches the absorption spectrum of previously reported ZnO NPs, as mentioned in Kayani et al. [31]. The spectrum is also in agreement with those obtained for ZnO NPs produced by synthetic methods [32,33,34,35,36,37], exhibiting a sharp, intense peak in the UV region at 376 nm and a small number of little, extreme, sharp peaks in between 250–350 nm. This is the characteristic absorption peak of ZnO. The absorption peak obtained may be either due to the surface plasmon resonance (SPR) involving the quantum size effect [38,39] or the semiconductor band gap transition [27]. Moreover, a small number of weak intense sharp peaks between 250–350 nm indicates the surface defects due to the recombination of electrons in the conduction band and holes in the valence band, which also specify the monodisperse nature of ZnO NPs distribution [27]. The optical bandgap of bio-synthesized ZnO NPs can be calculated from the Tauc plot, and it was estimated to be 3.24 eV.

#### 3.2.4. FESEM-EDX Analysis

The FESEM analysis was performed to identify the surface morphology of synthesized ZnO NPs and is displayed in Figure 4a–d. The resulting images have a particle size in the nanometre range, observable at higher magnification. The ZnO NPs show aggregation to some extent under lower magnification due to weak physical force. On increasing the magnification, the particles were good enough and well separated, and it is concluded that the particles are in the nanometre range. The chemical composition was analyzed using EDX spectra (Figure 4e), which showed four absorption peaks identified as Zinc (54.21%) and Oxygen (45.79%) in their atomic percentage, as well as Zinc (82.87%) and Oxygen (17.13%) in their weight percentage (Table 1), confirming the high purity nature of synthesized ZnO NPs.

#### 3.2.5. TEM and HR-TEM Analysis

ZnO NPs were analyzed by TEM analysis to understand the nanoparticles’ morphology further, this analysis is shown in Figure 5. Figure 5a shows that the ZnO NPs showed polygonal morphology and were non-uniform in size. The NPs also appeared to be monodispersed with an excellent crystalline structure, and agreed with the XRD results. The high-resolution TEM images (shown in Figure 5b) show clear lattice fringes without any distortion, designating the high crystallinity of ZnO NPs. The estimated interplanar spacing of adjacent lattice fringes is estimated to be 0.24 nm. Selected area electron diffraction (SAED) patterns of the sample have been captured and displayed in Figure 5c. The rings labelled in the SAED pattern represent the lattice planes and are identical to that known for the hexagonal wurtzite crystallite structure. To check the sample size distribution, 100 NPs were selected from different sample images. The particle size distribution of the sample is acquired from the average diameters, and the obtained histogram is shown in Figure 5d. The estimated average particle sizes of the sample were found to be 35 to 40 nm. The particle size determined from the TEM study is keenly matched with the crystallite size of ZnO acquired via XRD analysis.

#### 3.2.6. Raman Spectroscopy

The micro-Raman spectra, along with peak fitting using the Gaussian function for ZnO NPs in the spectral range 60–700 cm^−1^ is shown in Figure 6. The hexagonal wurtzite structure of ZnO can be represented by A_1_ + 2B_1_ + E_1_ + 2E_2_, as it belongs to the C6v point group. The polar modes, A_1_ and E_1_, can be split into transverse optical (TO) and longitudinal optical (LO) modes. B_1_ modes are Raman inactive. Raman active E_2_ modes are non-polar, comprise of E_2 (High)_ (437 cm^−1^) and E_2 (Low)_ (99 cm^−1^), which correlated with the vibration of the oxygen atom and heavier Zn-atom, respectively. The Raman bands appear at 332 and 380 cm^−1^, and are assigned to the 2E_2(M)_ and A_1_(TO) mode, respectively. The peak at 156 cm^−1^ may be related to a defect induced mode. The peaks at 284 and 580 cm^−1^ are attributed to B_1(Low)_ and B_1(High)_ modes, respectively [40]. The blurring of peaks can be seen in the spectra, suggesting disorders present in ZnO.

#### 3.2.7. XPS Analysis

The XPS spectra of ZnO was measured to explore the chemical states of the elements present in the sample. The broad range XPS spectrum of the sample is manifested in Figure 7a, and the high-resolution spectra of Zn 2p and O 1s were recorded and are depicted in Figure 7b,c, along with the deconvolution. All measured values are given with respect to the C 1s peak located at 284.6 eV. Figure 7b, the high-resolution XPS spectrum of Zn 2p, shows two strong peaks appearing at 1021.5 and 1044.6 eV, which are ascribed to the binding energies of Zn 2p3/2 and Zn 2p1/2, respectively. These values are in agreement with the previous reports on Zn^2+^ ion [41]. The spin−orbit splitting energy for the Zn 2p peaks was around 23.1 eV. The high-resolution O 1s spectra show an asymmetric peak, and the peak is fitted with two peaks centred at 530.2 eV and 531.5 eV, and correspond to the lattice oxygen of ZnO and hydroxyl group (–OH) oxygen.

### 3.3. Mechanism of Photo-Degradability of MB by ZnO Catalyst

The photocatalytic potency of green-synthesized ZnO NPs was illustrated with an example of MB dye under solar irradiation at an average intensity of 100,000 Lux (Figure 8). The MB photodegradation occurs due to the electrons, holes, and free radicals generated, and undergoes specific reactions to destroy the structure of MB. The catalyst, upon irradiation with sunlight, generates electrons in the conduction band and holes in the valence band. The generated electrons react with adsorbed oxygen on the catalyst to give O_2_^·−^ radical anion, and the water from the solution combines with the holes to produce OH· radicals. These generated radicals undergo the reduction and oxidation reaction while reacting with dyes [32,33,34,35,36,37]. However, O_2_^·−^ radical anion reacts further to generate HOO·, OH^−^ radicals, and OH^−^ anions, and these active species of oxygen are responsible for the degradation of MB dye.

As a result, the prime factor in the degradation of the dye is mainly originated from the generation of electrons and holes. These electron–hole pairs react further to generate active oxygen species radicals (O_2_^·−^, HOO^·^, OH^·^, and OH^−^), which are responsible for destroying the structure of MB dye and transforming it into a less harmful or degraded product. Hence, pollutant content in the water bodies can be minimized by using such a catalyst [42,43]. The photocatalytic reaction taking place within the ZnO catalyst can be summarised as shown below,
$\mathrm{ZnO}\;\left(\mathrm{catalyst}\right)+\;h\nu \;\left(\mathrm{light}\right)\to {\;e}^{-}\;\left(\mathrm{conduction}\;\mathrm{band}\right)+{\;h}^{+}\;\left(\mathrm{valence}\;\mathrm{band}\right)$$O_{2}+e^{-}\to {O_{2}}^{\cdot -}\;\left(\mathrm{radical}\;\mathrm{anion}\right)$$H_{2}O+h^{+}\to H^{+}+\mathrm{OH}\cdot \;\left(\mathrm{hydroxyl}\;\mathrm{radical}\right)$${O_{2}}^{\cdot -}+\mathrm{MB}\;\mathrm{dye}\to \mathrm{dye}\;\left(\mathrm{reduction}\right)$OH·+MB dye→dye (oxidation)

Further reaction,
$${O_{2}}^{\cdot -}+H^{+}\;\to \mathrm{HOO}\cdot$$
$$\mathrm{HOO}\cdot +e^{-}+H^{+}\;\to H_{2}O_{2}$$
$$H_{2}O_{2}+e^{-}\to \mathrm{OH}\cdot +\mathrm{OH}\cdot$$

### 3.4. Effect of Time of Solar Irradiation on Photodegradation of MB

The influence of the extent of solar irradiation on the percentage dye degradation of MB, evaluated by loading ZnO catalyst in the presence and absence of sunlight, is shown in Figure 9, which provides the absorption band for each sample at a definite regular time interval. The dye exhibited a characteristic absorption, with maximum absorbance at 664 nm in UV-visible spectra. By analyzing the absorption of the mixture for overall time, the efficiency calculated was found to be 98.07% upon solar irradiation and 8.57% without solar irradiation. This result implied that the biosynthesized ZnO photocatalyst exhibited an excellent photocatalytic activity only in the presence of sunlight.

### 3.5. Effect of ZnO Catalyst Loading on Photodegradation of MB

The impact of the catalyst concentration on photocatalytic degradation of MB dye was investigated by adding 5–15 mg/50 mL of ZnO catalyst into the MB solution, and the UV-visible spectra of the resultant solution are presented in Figure 10. The catalyst concentration has a noticeable impact on MB degradation, as evidenced by the graph.

The percentage of degradation of MB dye increased from 93.89% to 98.07% as the catalytic dosage increased from 5 mg to 15 mg. The effect of increment in the percentage degradation is probably due to more catalyst active sites and a higher adsorption area along with a higher specific surface area [44,45]. As a result of surface defects, a higher specific surface area is available for the origination of active radicals to degrade the dye. However, only a 5% increase in the photocatalytic efficiency was obtained with a higher amount of catalyst. This may be associated with the aggregation of particles, which increases the turbidity of the suspension, resulting from the high doses of the photocatalyst [46]. Because of it, the penetration of sunlight into the solution decreases, which results in a reduction in the photoactivated volume of suspension. Briefly, we can state that an increase in the catalyst concentration results in a higher adsorption area (more active site) as well as the generation of turbidity in the reaction mixture. Consequently, the photocatalytic efficiency concerning the amount of catalyst will be a combined impression of specific surface area and the solar light penetration. Hence, if one factor dominates over the other, the ultimate result will be produced in the extent of photocatalytic degradation of the MB dye.

### 3.6. The Effect of Calcination Temperature on the Photodegradation of MB

The relation between the calcination temperature of the ZnO photocatalyst and photocatalytic efficiency towards the MB degradation is shown in Figure 11. The percentage of degradation was found to be 82.41%, 87.45%, and 93.87% for the ZnO sample calcined at 300 °C, 500 °C, and 700 °C, respectively; the calcination temperature of the catalyst highly influenced its photodegradation capability. The increment of efficiency may be attributed to surface crystallinity, which relates to the specific crystallite area of the catalyst. The highest photocatalytic activity is observed for catalyst with a higher specific surface area at an elevated temperature, and it may be ascribed to more photogeneration of holes (Vb) and electrons (Cb) in the ZnO crystal, contributing to more defects as well [47,48]. The surface imperfections present in the ZnO lattice may evolve into the formation of small pores, which causes an increment in the surface area. These tiny pores benefit oxygen adsorption and thereby enhances the MB degradation rate. Hence, the higher specific surface area with an increasing calcination temperature helps in improved adsorption of MB dye on the ZnO surface and thus performs as a better photocatalyst for photodegradation of MB.

### 3.7. Effect of Precursor Concentration on Photocatalytic Efficiency

The effect of the concentration of the precursor on the photocatalytic efficiency of the photocatalyst is presented in Figure 12. The photodegradation ability of the ZnO catalyst is in a linear relation with the Zn(NO_3_)_2_·6H_2_O concentration, which was utilized in the synthesis process and attained a maximum of 87.45% for 0.1 M. The higher degradation capability can be ascribed to the variation in average crystallite size of the synthesized ZnO nanostructure. The crystallite size calculated from the XRD data shows that the average crystallite size of ZnO increases in the precursor concentration. A detailed description is given in Section 3.8. The increase in crystallite size results in a higher available surface area for the adsorption of more oxygen and photogeneration of more electrons and holes [41,46], i.e., if the crystallite size is increasing, crystallinity may increase. This may be due to the agglomeration of crystallite size particles which may result in the long-range of orderly arrangement of particles. Moreover, the crystallinity and larger crystallite size may enhance the reduction in the bandgap, and may generate more e-h pairs upon irradiation with solar light, which helps in producing radicals. Hence, the increase in the precursor concentration benefits the photodegradation of dyes. The photocatalytic activity depends on several factors, including surface area, particle size, crystallinity, defect (pores), adsorption (active sites), catalyst concentration, and calcination temperature. It does not rely on one factor; dominance of one factor over the other will decide the photocatalytic degradation of dyes by the ZnO photocatalyst.

### 3.8. The Effect of Concentration of Zn(NO_3_)_2_ on Crystallite Size of ZnO NPs

To study the effect of Zn(NO_3_)_2_ concentration on the average crystallite size, ZnO NPs were synthesized from three different concentrations of Zn(NO_3_)_2_·6H_2_O solution (0.025 M, 0.05 M, and 0.1 M) and the samples were calcined at 500 °C. The XRD spectra of ZnO NPs synthesized from the previously mentioned concentration are depicted in Figure 13. The Debye–Scherrer equation was used to determine the average crystallite size of green ZnO from the XRD pattern. It was observed that the intensity of every diffraction plane is increasing, but the peak of the (101) plane is of particular interest since it is the most intense peak. Hence, the crystallite size calculations were done with the (101) diffraction peak using the Scherrer formula.

The increase in precursor concentration leads to the enhancement in the preferential orientation of the intense peak, which specified the improvement in crystallinity [35,36]. The crystallite size found a maximum for 0.1 M precursor solution, which is around 49.25 nm. Similarly, the crystallite size obtained for 0.05 M and 0.025 M are 30.44 nm and 25.51 nm, respectively. That the average crystallite size decreases with a decrease in concentration is due to the increment in the FWHM value of (101) diffraction peak, as evidenced in Table 2. The growth in the crystallite size is due to the crystal agglomeration with an increase in the concentration.

Moreover, in a highly concentrated solution, the rate of reduction and oxidation of Zn(NO_3_)_2_ to Zn(OH)_2_ will be higher, as the collision between highly concentrated salt with phytochemicals will be more. Therefore, the chance of lower concentration reduction will be minimal as there are a small number of salt ions present. This increase in crystallite size is also reflected in photocatalytic studies, as discussed earlier, in which the photocatalytic activity is enhanced with the increase in precursor concentration.

The photocatalytic efficiency of *C. tamala* stabilized ZnO NPs is compared with recent studies from the literature and is shown in Table 3.

## 4. Conclusions

The green synthesis of ZnO NPs using the *C. tamala* template offers a simple, stable, sustainable, economic, and eco-friendly approach compared to other conventional methods that have been commonly deployed. The phytochemicals present in the *C. tamala* extract played a significant role in serving both as capping and reducing agents during the synthesis. The synthesized ZnO NPs were hexagonal crystalline in nature, having an average size of 35 nm. The bandgap obtained from the Tauc plot was 3.24 eV. It is noticeable that the photocatalytic activity of *C. tamala* stabilized ZnO NPs was exclusively dependent on various factors and the conditions optimized for the best catalytic performance. It was shown to have higher photocatalytic activity for the catalyst loading, 5 mg/50 mL. ZnO synthesized from 0.1 M Zn(NO_3_)_2_·6H_2_O and ensuing ZnO calcined at 700 °C displayed higher photocatalytic efficiency towards MB degradation. The ZnO microstructure formation process and the crystallite size have a prominent impact on the photocatalytic performance of ZnO NPs. The present study has outstandingly suggested that green synthesized ZnO NPs display an excellent photocatalytic degradation capability of a maximum of 98% towards the degradation of organic dye. Hence, the presently adopted method provides a better alternative over other methods and can be developed for a large-scale operation to be employed in wastewater treatment and water purification.

## Figures and Tables

**Figure 1 nanomaterials-11-01558-f001:**
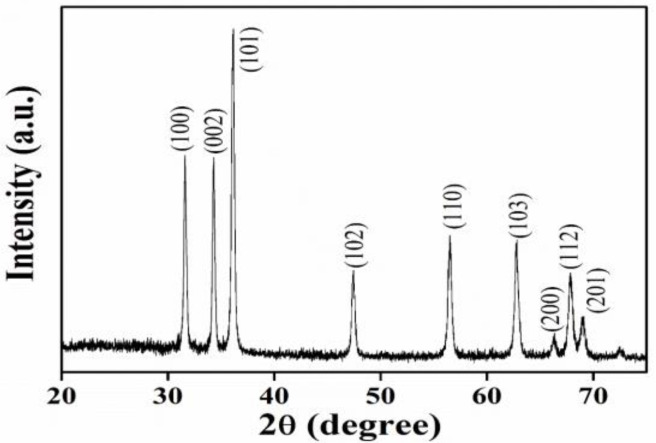
XRD spectrum of ZnO NPs.

**Figure 2 nanomaterials-11-01558-f002:**
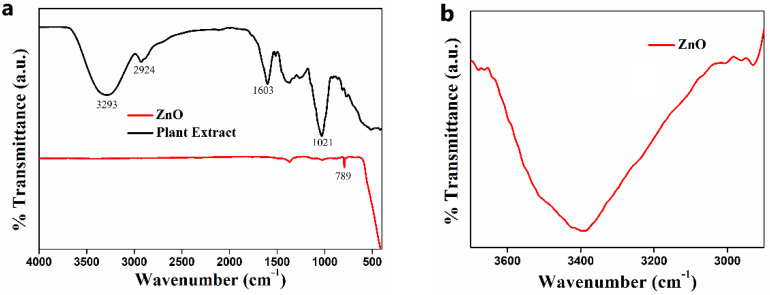
(**a**) FT-IR spectra of leaf extract and ZnO NPs; (**b**) The expanded region of FT-IR spectra around 3100 to 3500 cm^−1^ of the ZnO sample.

**Figure 3 nanomaterials-11-01558-f003:**
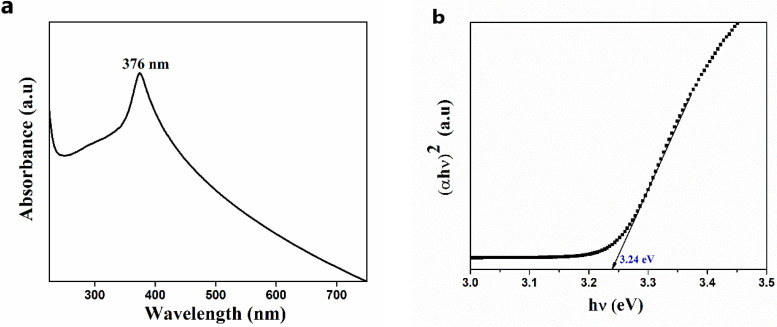
(**a**) The absorption spectrum of ZnO NPs; (**b**) Tauc plot of ZnO NPs.

**Figure 4 nanomaterials-11-01558-f004:**
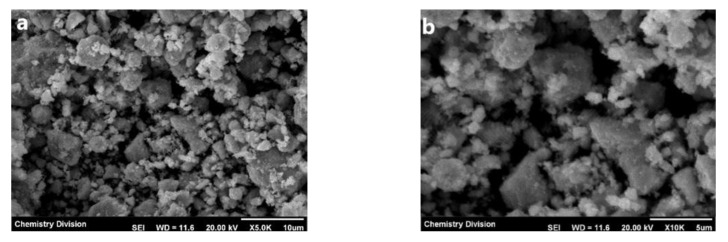
Field Emission Scanning electron microscopy (FESEM) images of ZnO NPs in magnification: (**a**) ×5.0 k; (**b**) ×10 k; (**c**) ×15 k; (**d**) ×30 k; (**e**) EDX (Energy Dispersive X-ray) Analysis of ZnO NPs.

**Figure 5 nanomaterials-11-01558-f005:**
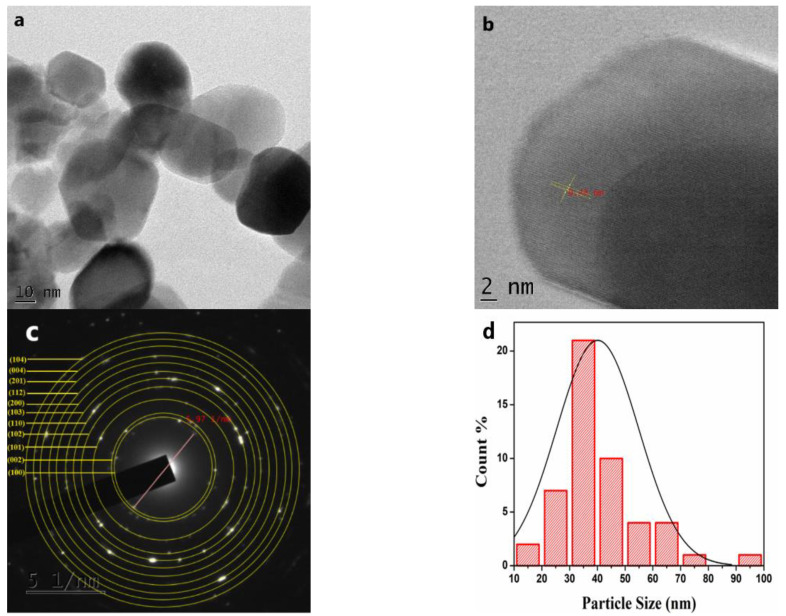
TEM data of ZnO NPs: (**a**)TEM image; (**b**) HR-TEM image; (**c**) SAED pattern; (**d**) Histogram showing particle size distribution in the sample.

**Figure 6 nanomaterials-11-01558-f006:**
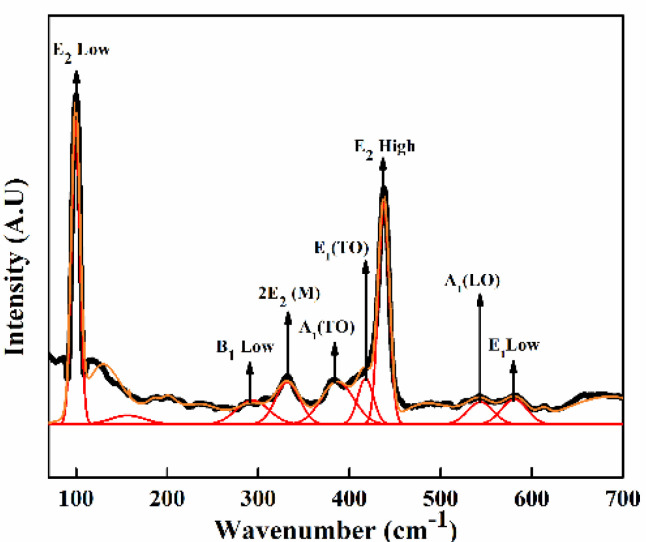
Raman spectrum of ZnO NPs.

**Figure 7 nanomaterials-11-01558-f007:**
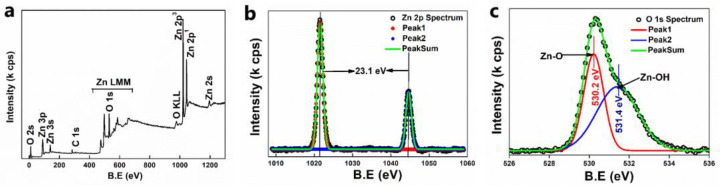
(**a**) Wide range XPS spectra of ZnO NPs; (**b**) high-resolution spectra of Zn 2p; (**c**) high-resolution spectra of O1s.

**Figure 8 nanomaterials-11-01558-f008:**
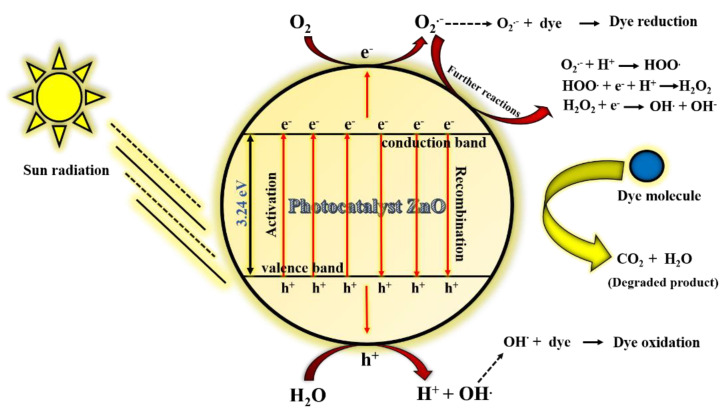
Schematic diagram of photocatalysis mechanism of ZnO NPs.

**Figure 9 nanomaterials-11-01558-f009:**
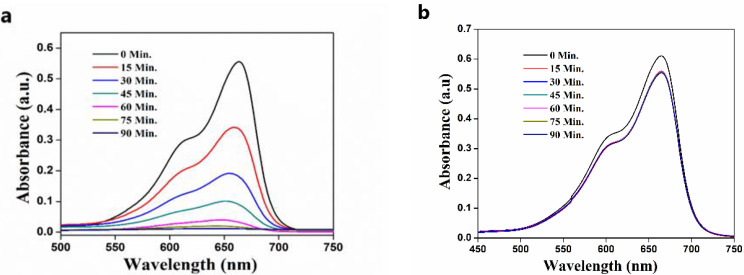
The photo-degradability of MB dye by ZnO catalyst (**a**) in the presence of sunlight; (**b**) in the absence of sunlight.

**Figure 10 nanomaterials-11-01558-f010:**
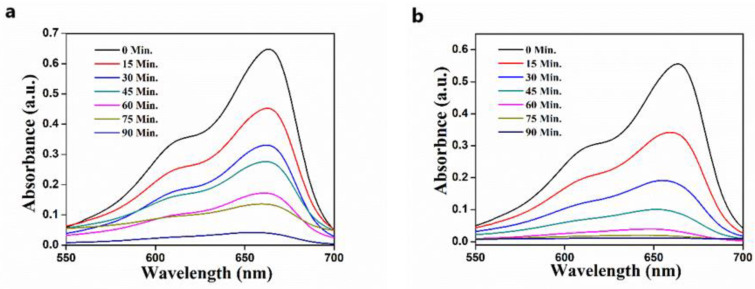
The effect of ZnO photocatalyst on % degradation of MB dye (**a**) with 5 mg ZnO; (**b**) with 15 mg ZnO.

**Figure 11 nanomaterials-11-01558-f011:**
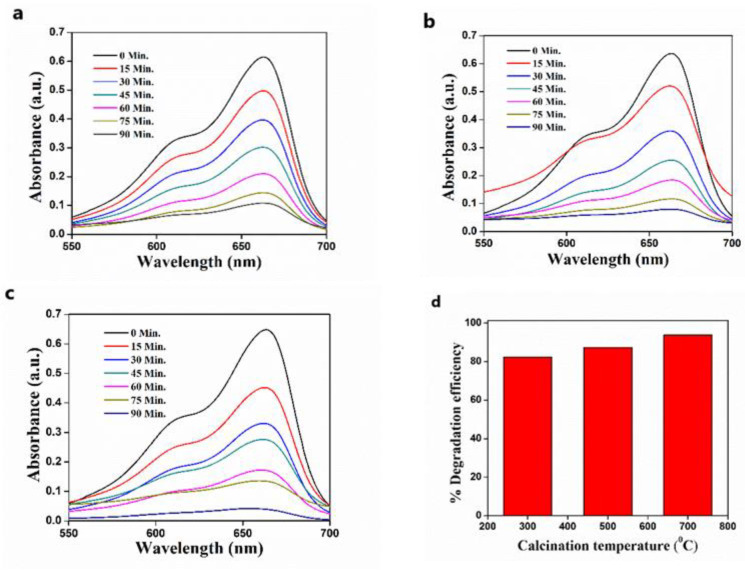
The photocatalytic degradation of MB after treatment with ZnO calcined at (**a**) 300 °C; (**b**) 500 °C; (**c**) 700 °C; (**d**) the effect of calcination temperature of ZnO on MB photodegradation.

**Figure 12 nanomaterials-11-01558-f012:**
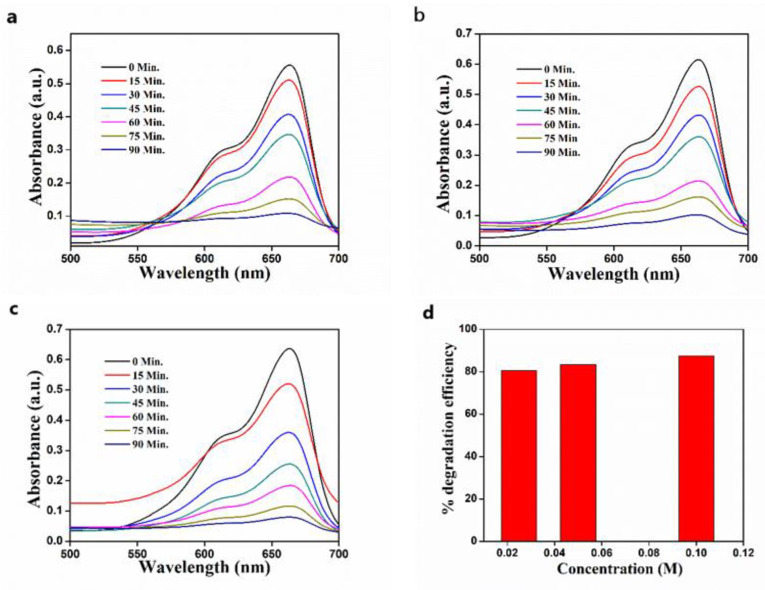
The photocatalytic degradation of MB after treatment with ZnO synthesized from (**a**) 0.025 M Zn(NO_3_)_2_; (**b**) 0.05 M Zn(NO_3_)_2_; (**c**) 0.1 M Zn(NO_3_)_2_; (**d**) the effect of Zn(NO_3_)_2_·6H_2_O concentration on photodegradation of MB.

**Figure 13 nanomaterials-11-01558-f013:**
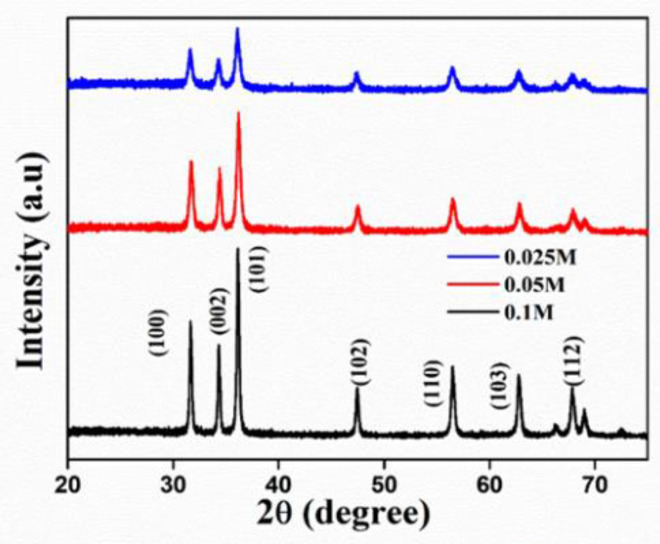
XRD patterns of ZnO NPs synthesized from the different molar concentrations of precursor.

**Table 1 nanomaterials-11-01558-t001:** Elemental Composition of ZnO.

Element	Weight%	Atomic%
Zn	82.87	54.21
O	17.13	45.79
Total	100	100

**Table 2 nanomaterials-11-01558-t002:** Crystallite size of ZnO NPs by varying Zn(NO_3_)_2_ concentration.

Sl. No.	Concentration (M)	2Ɵ (Degree)	FWHM	Crystallite Size (nm)
1	0.025	36.28	0.56569	25.51
2	0.05	36.20	0.47367	30.44
3	0.1	36.15	0.28872	49.95

**Table 3 nanomaterials-11-01558-t003:** A comparative table showing the photocatalytic efficiency of *C. tamala* stabilized ZnO NPs with other phtotcatalytic system based on ZnO NPs for the removal of MB.

Sl. No.	ZnO Synthesis Method	% Efficiency (Time)	Light Source	MB Concentration	Reference
1	Precipitation Sol-gel Method	81% (180 min) 92.5% (180 min)	UV light	20 mg/L	[49]
2	Sol-gel	86% (140 min)	UV light	10 mg/L	[50]
3	Green synthesis-*Becium grandiflorum* aqueous leaf extract	69% (200 min)	UV light	100 ppm	[51]
4	Green synthesis-Durian waste	84% (40 min)	Solar light	10mg/L	[52]
5	Ultra sound assisted green synthesis	93.25% (70 min)	Solar light	1 mg/mL	[53]
6	Green synthesis-*Kalopanax septemlobus*	97.5% (30 min)	UV light	50 µM	[54]
7	Green synthesis-*Sambucus ebulus*	80% (200 min)	UV light	50 ppm	[55]
8	Green synthesis-*Cinnamomum tamala* aqueous leaf extract	98% (90 min)	Solar light	10 µM	Present study

## Data Availability

The data presented in this study are available on request from the corresponding author.

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
