# Peer review of "Cinnamomum tamala Leaf Extract Stabilized Zinc Oxide Nanoparticles: A Promising Photocatalyst for Methylene Blue Degradation"

_nanomaterials, 2021, doi:10.3390/nano11061558_

Round 1
Reviewer 1 Report
Comments are included in the attached file

Reviewer 2 Report
ZnO nanoparticles with a hexagonal wurtzite crystallite structure were synthesized by a plant extract-mediated approach as a promising photocatalyst for photodegradation of methylene blue. Cinnamomum tamala leaves extract was used as a reducing and capping agent in this case. The developed materials were characterized by a wide spectrum of various techniques (including XRD, HR-TEM, SEM, XPS, UV-Vis, FT-IR and Raman spectroscopy) which allowed to recognize their properties. Generally, I find the work interesting. Therefore, the manuscript may be considered for publication in the Nanomaterials journal, but the following issues should be clarified.
- How do the authors rate the effectiveness of the extraction of active ingredients from Cinnamomum tamala leaves in pure water?
- How were the samples prepared before the microscopic measurements? In addition, it should be indicated in the XPS description whether sample neutralization was used and which regions were measured.
- On the basis of intensity of which UV-Vis band the concentration of dye in the solution subjected to photodegradation was determined?
- Please analyze whether withdrawing 5 ml samples for spectrophotometric analyzes did not disturb equilibrium in the solution.
- How does increasing the calcination temperature affect the size of ZnO crystallites and their aggregation?
- Verify the FT-IR peak position given in line 234.
- The authors can consider specifying magnitude of SEM images presented in Fig. 4a-d in the figure caption.
- Can it be assumed that the effect of removing methylene blue without solar irradiation can be attributed to adsorption on ZnO nanograins?
- Changes in photoactivity of the ZnO samples calcined at different temperatures are discussed in relation to their porosity. Meanwhile, the textural parameters of these materials were not determined.
- The transformation of Zn(NO3)2 to Zn(OH)2 described in lines 439/440 cannot be called reduction, because then there is no change in the oxidation state of zinc.
- All typos should be eliminated from the manuscript.
Author Response
"Please see the attachment."

Reviewer 3 Report
In this work, "Cinnamomum tamala leaf extract stabilized zinc oxide nanoparticles: A promising photocatalyst for methylene blue degradation", the authors proposed a facile green synthetic method for the synthesis of ZnO nanoparticles. Based on the obtained results, the authors claimed that the proposed for ZnO synthesis shows promising photocatalytic capability over the dye molecules. Overall, this manuscript has a strong potential for a second review after applying the issues and addressing the shortcomings listed below:
1-The authors should polish/revise some grammatical mistakes and typos along the manuscript. I invite the authors to read their manuscript carefully and make the required changes where necessary.
2-In the Introduction section, while discussing recent developments in the field of ZnO nanoparticles, the following works should be considered and cited to give a more general view to the possible readers of the work: [(i) Controlled self-assembly of plasmon-based photonic nanocrystals for high performance photonic technologies, Nano Today 37, 101072 (2021); (ii) Sonochemical synthesis of a zinc oxide core-shell nanorod radial p-n homojunction ultraviolet photodetector, ACS Applied Materials & Interfaces 9, 19791 (2017)].
3-Corresponding references should be provided if the equations are taken from some other study.
4-In Figure 2, you can plot the inset as Figure 2b, on the right side of the original Figure 2 (in its current form, the inset is hard to read).
5-In Figure 9b, you can remove the frame around the legend.
Round 2
Reviewer 1 Report
Most of the more important issues directed to the authors have been suitably discussed. In my oppinion the revised paper is publishable now.
Reviewer 3 Report
In its current form, the revised manuscript is suitable for publication.